# How Does Procedural Justice Affect Job Crafting? The Role of Organizational Psychological Ownership and High-Performance Work Systems

**DOI:** 10.3390/bs15010004

**Published:** 2024-12-24

**Authors:** Zhun Gong, Mengxuan Ren, Yingjie Sun, Ziyi Zhang, Wen Zhou, Xiaowei Chen

**Affiliations:** 1Department of Psychology, School of Education Science, Qingdao University, Qingdao 266075, China; gongzhun2001@qdu.edu.cn (Z.G.); rmx19970128@163.com (M.R.); 15606423072@163.com (Y.S.); zhangziyiazi@163.com (Z.Z.); wenzhou0923@163.com (W.Z.); 2Institute for Educational Measurement and Evaluation, Qingdao University, Qingdao 266075, China; 3Academic Affairs Office, Zhejiang Shuren University, Hangzhou 310015, China

**Keywords:** procedural justice, job crafting, organizational psychological ownership, high-performance work systems, hierarchical linear modeling

## Abstract

In today’s highly competitive and complex market environment, enhancing adaptability has become essential for the sustainable development of enterprises. Job crafting, an important strategy for strengthening a company’s core competitiveness, has garnered increasing attention in recent years. However, previous studies have often overlooked its antecedent variables and mechanisms. This study draws on social exchange theory and equity theory to examine how procedural justice influences the cross-level pathways of job crafting. Hierarchical linear modeling (HLM) was employed to analyze data from 76 companies and 1049 employees. The results demonstrate that procedural justice significantly and positively impacts employee job crafting. Additionally, organizational psychological ownership mediates the relationship between procedural justice and job crafting at a cross-level, while high-performance work systems positively moderate the link between organizational psychological ownership and job crafting. These findings reveal a novel pathway for enhancing employee job crafting and offer practical insights for corporate management. Companies should focus on fostering an environment characterized by procedural justice and which promotes organizational psychological ownership to encourage job-crafting behaviors. Moreover, attention should be given to the effectiveness of organizational psychological ownership and to the moderating role of high-performance work systems in this process.

## 1. Introduction

With rapid changes in both the internal and external environments of firms and the increasing qualifications of the workforce, the limitations of the traditional top-down approach to job design in organizational management have become more evident. This traditional approach often overlooks the importance of leveraging employees’ initiative and fails to meet their expectations and needs fully ([3]). As a result, promoting employee involvement in job design has become increasingly crucial. Job crafting, introduced by [36] ([36]), is a bottom-up design approach that emphasizes employee motivation and autonomy. It enables organizations to better address employees’ needs while enhancing their sense of engagement and belonging. With the rise in the boundaryless career concept, a new contractual relationship between individuals and teams has been established, and individual job crafting can strengthen the sense of team value, improve person-team fit, and increase job satisfaction and performance ([11]). As organizations adapt to changing environments, they increasingly rely on employees to take initiative in job crafting ([10]). Therefore, motivating employees to redesign their work has become a central theme in organizational management research.

Current research on job crafting, however, primarily focuses on individual job attitudes and outcomes, such as increased job satisfaction ([11]; [33]), enhanced work engagement ([28]), and improved organizational performance ([33]). There is relatively little research on the antecedent variables of job crafting, particularly the role of situational factors. Contextual factors are essential for understanding employees’ job crafting because job crafting positively impacts both employees and organizations. It is critical for organizations to understand what kind of environment fosters more effective job-crafting behaviors ([27]). Therefore, this study examines the mechanisms shaping job crafting from the contextual perspective of business organizations, aiming to assist companies in improving their management practices.

Procedural justice, as part of a supportive organizational environment, involves adherence to fair procedural standards. When valued, it can significantly improve job performance, employee behavior, job satisfaction, and work attitudes ([13]). Drawing on social exchange theory and equity theory, employees tend to respond positively to organizations when their efforts benefit them. This suggests that an organization’s ability to create an equitable environment enhances employees’ psychological security, boosts self-efficacy and motivation, and promotes engagement and job-crafting behaviors ([14]). Procedural justice operates at both the organizational and individual levels, with organizational-level procedural justice referring to the actual climate of fairness created by the organization, and individual-level procedural justice referring to employees’ perceptions of their treatment ([29]). Therefore, procedural justice at the organizational level is more suitable to be discussed in this study as an organizational situational factor, specifically as an antecedent variable of employees’ job crafting. Accordingly, this study will explore the impact of procedural justice on employees’ job-crafting behaviors and its underlying mechanisms at the organizational level.

To investigate the relationship between procedural justice and employee job crafting, this study focuses on uncovering the mechanisms mediating this relationship. Organizational psychological ownership refers to employees’ sense of ownership over the organization. Previous research has shown that a fair organizational environment is an antecedent of organizational psychological ownership, enhancing employee engagement and promoting positive behaviors ([23]). Additionally, job crafting has been closely linked to organizational psychological ownership, with studies indicating that organizations can increase job crafting behaviors by fostering psychological ownership ([35]). Drawing on job demand-resource theory and social exchange theory, it is expected that organizations providing a fair work environment will enhance employees’ emotional attachment and psychological ownership, leading them to engage in job-crafting behaviors by viewing themselves as contributors to the organization’s success ([26]). Therefore, this study hypothesizes that procedural justice can enhance employees’ psychological ownership, motivating them to engage in job-crafting behaviors.

In addition, when organizations establish fair working environments, employees are more willing to engage in job crafting to enhance the meaning of their work. This dynamic interaction between employees and the organizational context leads to the transformation of human resource (HR) practices into work outcomes ([25]). To better understand how procedural justice enhances job crafting, it is also crucial to consider the role of corporate HRM practices. High-performance work systems (HPWSs) are a set of internally aligned and externally consistent HRM practices with organizational strategy ([21]). [4] ([4]) argue that the organizational context significantly influences the implementation of HR practices and that procedural justice constitutes a critical context in the HPWS performance relationship. HPWSs are often thought to fulfill employees’ psychological needs. According to self-determination theory, when employees’ basic psychological needs are satisfied, they are motivated to be more autonomous and to feel a sense of belonging to the company. This motivation can, in turn, influence employees’ job crafting and the role of their psychological ownership within the organization ([21]). Therefore, the third objective of this study was to explore the moderating role of HPWSs on the mediating effect of organizational psychological ownership between procedural justice and job crafting.

In summary, this study makes several contributions to job crafting. Firstly, unlike previous studies that focused on outcome variables or individual aspects, we adopt an organizational contextual perspective, emphasizing organizational-level contextual factors. We examine the mediating role of organizational psychological ownership between procedural justice and employees’ job-crafting behaviors using a cross-level model of moderating mediators, and also examine how organizational psychological ownership influences the relationship between job-crafting behaviors under varying levels of high-performance work system implementations.

## 2. Theoretical Framework

### 2.1. Procedural Justice and Job Crafting

Past research has proposed procedural justice as a key factor influencing employees’ work behaviors; when organizations follow fair relational norms in decision-making, it leads to further identification with the organization, motivating positive work behaviors, and, ultimately, voluntary service to the organization ([19]). Job crafting, on the other hand, is an employee’s initiative to adapt what and how he or she does his or her job to fit the needs of the organization, and it is a dimension for evaluating positive work behaviors ([36]; [33]). Thus, procedural justice is potentially linked to job crafting. However, in the past, researchers have mainly focused on the effects of job crafting on the positive psychological cognitive and affective aspects of individuals ([30]), as well as on work attitudes and work outcomes ([18]), for example, employee-authored job crafting behaviors positively affect employees’ health ([32]) and job satisfaction ([6]). However, less attention has been paid to how organizational contextual factors, such as procedural justice, as antecedent variables, affect job crafting.

The relationship between procedural justice and job crafting can be explained by combining equity and social exchange theories. Equity theory suggests that employee motivation is a function of perceptions and judgments based on comparisons of fairness to other criteria ([1]), while social exchange theory emphasizes the concept that interpersonal relationships in organizations are based on the process of cost–benefit exchanges, whereby each individual receives the resources and support he or she needs by giving and receiving in return ([8]). Therefore, it can be argued that when employees perceive that the organization is treating them fairly, their motivation to work increases and they receive more resources by giving back to the organization. Previous research has shown that when procedural justice is high in an organization, it fosters trusting relationships within the organization and allows for employees to feel that their value is recognized, which increases their sense of dependence on the organization and their motivation, willingness, and responsibility to engage in positive job crafting ([31]).

Based on this study, we propose that procedural justice in an organization affects employees’ job-crafting behavior, and the following hypotheses are formulated:

**Hypothesis** **1:**
*Procedural justice is a significant positive predictor of employee job crafting.*


### 2.2. The Mediating Role of Psychological Ownership of Organizations

Organizational psychological ownership is an attitude in which the employee’s sense of possession of the organization influences the employee’s affective identification and commitment to the organization ([5]). Job demand-resource theory suggests that employees’ psychological states are antecedents of employees’ work outcomes. When organizations establish psychological ownership with their employees, employees develop more positive attitudes toward the organization and increase their motivation and initiative at work ([26]). Thus, organizational psychological ownership may provide a better explanation for positive behaviors in the workplace, such as employees’ job-crafting behaviors ([13]). According to social exchange theory, procedural justice at the organizational level can bring about a fair and just supportive organizational environment which can make employees feel the responsibility of ownership and inspire them to devote themselves to the company’s loyalty and willingness, thus increasing their organizational psychological ownership ([16]; [12]; [22]).

To sum up, combining job demand-resource theory and social exchange theory, we hypothesize that when the organization establishes a just and fair working environment for employees, employees will actively and positively adjust their work and increase job crafting behavior. That is, procedural justice at the organizational level will have an impact on employees’ organizational psychological ownership, which in turn affects employees’ job-crafting behaviors. Based on the above analysis, this study proposes the following hypotheses:

**Hypothesis** **2a:**
*Procedural justice is a cross-level positive predictor of employees’ psychological organizational ownership.*


**Hypothesis** **2b:**
*Employee organizational psychological ownership is a significant positive predictor of employee job crafting.*


**Hypothesis** **2c:**
*Organizational psychological ownership can assume a cross-level mediating role between procedural justice and employee job crafting.*


### 2.3. The Moderating Role of High-Performance Work Systems

[24] ([24]) argue that motivation-enhancing HR practices encourage employees to proactively engage in challenging job crafting, a critical aspect of job crafting. The high-performance work system (HPWS) is an integrated human resource management approach aimed at improving business performance by empowering, motivating, and providing employees with opportunities ([21]). In line with social exchange theory, HPWS can positively influence employees’ job-crafting behaviors by shaping their perceptions of procedural justice. When an organization effectively implements the HPWS, employees perceive the performance appraisal process as fair and are more likely to view job crafting as beneficial to both the organization and their own career development ([2]). Furthermore, from a self-determination perspective, HPWS fosters stronger socio-emotional exchanges based on trust. In a supportive environment, this enhances employees’ organizational identity and sense of belonging, which increases organizational psychological ownership. This heightened ownership subsequently leads to greater autonomous motivation to engage in job-crafting activities ([21]; [2]; [15]). Therefore, this study posits that high-performance work systems can influence both procedural justice and organizational psychological ownership. In summary, the following hypotheses are proposed in this study:

**Hypothesis** **3a:**
*High-performance work systems moderate the positive relationship between procedural justice and employee job crafting.*


**Hypothesis** **3b:**
*High-performance work systems moderate the positive relationship between procedural justice and employees’ organizational psychological ownership.*


**Hypothesis** **3c:**
*High-performance work systems can play a moderating role in the positive relationship between the psychological ownership of employees’ organizations and employees’ job crafting.*


The proposed theoretical model is illustrated in Figure 1.

## 3. Methodology

### 3.1. Research Methodology and Sample

This study employed a questionnaire survey method to examine 92 science and technology-based enterprises in eastern China, encompassing sectors such as new energy, advanced manufacturing, new materials, electronic information, and biotechnology, with involvement from various departments including technology, finance, and HR. We selected science and technology-based enterprises as the research subjects because these enterprises typically exhibit high innovation vitality and technological content, playing a crucial role in dynamic market environments. We screened enterprises that met these criteria by reviewing their registration information, industry reports, and public information. To ensure the sample’s representativeness, we selected enterprises ranging from small to large, with employee counts between 50 and 5000. We initially screened approximately 200 technology-based enterprises using industry reports and corporate databases. We then contacted the HR departments or management of these enterprises to explain the study’s purpose and methodology and to solicit their willingness to participate. Ultimately, 92 companies agreed to participate in our study. In each participating firm, we selected employees using a random sampling method. At least 10 employees were selected from each company to ensure a diverse and representative sample. In total, we collected survey data from 1345 employees. The questionnaire survey was conducted from 1 August 2021, to 31 December 2021, spanning three months. Before administering the questionnaire, we obtained permission from the enterprise leaders and the HR department. Employees were fully informed of the survey’s purpose and the confidentiality of the results. Each participant received a small gift to reduce stress after completing the survey. To control for homogeneous variance, this study used a multi-source data collection approach. The questionnaires were coded and numbered for each organization before distribution.HR executives evaluated the high-performance work system and basic firm information, while employees assessed job crafting, organizational psychological ownership, procedural justice, and personal information (e.g., gender, age, education level, tenure, and job title). The survey involved 92 technology-based enterprises and 1345 employees. Based on the criteria of polygraph question filling status, completeness of questionnaire filling, and length of answering time, 76 enterprises were finally retained as valid subjects after invalid answered questionnaires were excluded, and valid questionnaire data of 1049 subordinate employees were recovered, and the effective recovery rate of the questionnaires was 77.9%. The relevant demographic information is as follows. The results of descriptive statistics are shown in Table 1 and Table 2.

### 3.2. Measurement Tools

To ensure the reliability and validity of the measurement results of this study, we chose a well-established scale that has been widely validated both at home and abroad as a research tool. In this study, we used the Likert 5-point scale as the scale scoring method, where 1 stands for “not conforming at all” and 5 stands for “conforming completely”.

**Procedural justice scale:** Procedural justice was measured using the scale used in [7]’s ([7]) study. The scale consists of seven measurement items, such as “I can express my opinions and feelings during the decision-making process”. In this study, the internal consistency coefficient of the procedural justice scale was 0.899.

**Organizational psychological ownership scale:** Organizational psychological ownership was measured using a unidimensional 7-item scale developed by [34] ([34]). Typical items include “I feel a high degree of personal ownership in this organization”. The internal consistency coefficient for this scale in this study was 0.892.

**High-performance work system scale:** The high-performance work system was measured using the scale developed by [9] ([9]), which was validated in a Chinese context and then applied more broadly in this study. Typical questions include “Employees have a clear career development path in the organization”. In this study, the internal consistency reliability of the scale was 0.958.

**Job Crafting Scale:** The Job Crafting Scale uses four items from the Individual Job Crafting Scale designed by [20] ([20]) and adapts situational words to fit the organizational context, e.g., “I will fine-tune the workflow to improve efficiency”. In this study, the internal consistency coefficient of the scale was 0.875.

Specific questions on the above scales can be found in the Appendix A.

## 4. Results

### 4.1. Data Aggregation

Since procedural justice is an organizational-level variable, its evaluation is self-assessed by employees rather than organizational leaders. Therefore, the consistency of internal employee results needs to be considered, and, based on this, individual-level data need to be aggregated at the organizational level. At the same time, between-group differences in employee job crafting need to be considered to determine whether cross-level analyses are possible. In this study, we used the intragroup correlation coefficient ICC and the intragroup consistency coefficient rwg as the relevant calculation indexes to determine whether the data at the individual level can be aggregated to the organizational level and the intergroup differences in employee job crafting. The test results show that the ICC and rwg indicators can effectively assess these factors and help us determine whether data can be aggregated and analyzed at the organizational level. The test results show that the ICC of procedural justice is 0.639 > 0.138 and the rwg mean is 0.77, which is within a reasonable range, and the intra-group consistency of the variables is good, so procedural justice at the employee level can be aggregated to the organizational level. Meanwhile, the ICC of job crafting is 0.662 > 0.138 and the rwg mean is 0.76. According to the classification criteria of intragroup correlation coefficient ICC, 0.01 ≤ ICC ≤ 0.059 was categorized as low correlation, 0.059 ≤ ICC ≤ 0.138 was categorized as moderate correlation, and ICC ≥ 0.138 was categorized as high correlation. These categorization criteria can be used to assess the degree of association and provide a basis for quantitative analysis in empirical studies. According to this criterion, it indicates a high degree of intra-group correlation and significant inter-group differences, enabling two-level linear modeling statistical analysis using a multilayer linear model (HLM). In this study, SPSS 20.0 software and Mplus 8.0 software were used to statistically analyze all the data obtained from the measurements.

### 4.2. Common Method Bias Test

This study used the Harman one-way test to assess the effect of common method bias in the collected data. This method places all measures in an exploratory factor analysis model to detect common method bias effects. Through the unrotated factor analysis, we found that the first factor contributed 19.61% of the variance, which was below the critical criterion of 40%.

### 4.3. Structural Validity Tests

In this study, a validated factor analysis was conducted using the structural equation analysis tool in AMOS 21.0 software. The method assesses the validity of the measurement instrument by looking at the model fit of the sample data. The results of this study show that the results of the structural validity tests for the variables reported in Table 3 met acceptable standards. This indicates that the measures of the variables in the tests of this study have a good fit for the actual situation.

### 4.4. Descriptive Statistics and Correlation Analysis

The matrix of the results of the data analysis of the means, standard deviations, and correlation coefficients of the variables involved in this study is shown in Table 4. From Table 4, it can be concluded that procedural justice is significantly and positively correlated with job crafting (*r* = 0.85, *p* < 0.01), procedural justice is significantly and positively correlated with organizational psychological ownership (*r* = 0.29, *p* < 0.01), and organizational psychological ownership is significantly and positively correlated with job crafting (*r* = 0.21, *p* < 0.01), and there is a significant correlation between all variables that provided initial support for the subsequent testing of the hypotheses.

### 4.5. Hypothesis Testing

In this study, Mplus 8.0 was used to construct cross-level structural equation modeling to test all the hypotheses; the specific testing steps are as follows, and the results of the relevant test data are shown in Table 5.

The results of the direct effect test are shown in Model 1: procedural justice has a positive effect on employee job crafting (γ = 1.041, *p* < 0.01), and Hypothesis 1 is verified.

The mediation effects test, as shown in Table 5, showed that procedural justice had a cross-level positive effect on organizational psychological ownership (*γ* = 0.548, *p* < 0.01), and Hypothesis 2a was tested. Organizational psychological ownership had a significant positive effect on job crafting (γ = 0.106, *p* < 0.05), and Hypothesis 2b was tested. The value of the mediating effect of organizational psychological ownership between procedural justice and job crafting is equal to the value of the effect of procedural justice on organizational psychological ownership multiplied by the value of the effect of organizational psychological ownership on job crafting, i.e., 0.548 × 0.106 = 0.058, *p* < 0.05, with 95% confidence interval [0.016, 0.100], and Hypothesis 2c was verified.

Moderating effect test. As shown in Table 5, the interaction term between procedural justice and high-performance work systems had a non-significant effect on job crafting (γ = 0.061, *p* > 0.05), i.e., the moderating effect of high-performance work systems between procedural justice and job crafting was not significant, and Hypothesis 3a was not valid. The interaction term between procedural justice and high-performance work systems was not significant (γ = 0.005, *p* > 0.05) on organizational psychological ownership, i.e., the moderating role of high-performance work systems between procedural justice and organizational psychological ownership was not significant, and Hypothesis 3b was not valid. The interaction term between organizational psychological ownership and the high-performance work system has a significant effect on job crafting (γ = 0.193, *p* < 0.05) with a 95% confidence interval of [0.055, 0.332]; thus, the moderating effect of the high-performance work system between organizational psychological ownership and job crafting is significant and Hypothesis 3c is supported. In order to more visually reflect the moderating effect of high-performance work systems (HPWS), this study conducted a simple slope analysis to plot the moderating effect of high-performance work systems (HPWS) on the relationship between organizational psychological ownership and job crafting at levels one standard deviation above and below the mean. As can be seen in Figure 2, at lower levels of HPWS (mean − 1 standard deviation), the effect of organizational psychological ownership on job crafting was not significant, with a simple slope *b* = −0.104, *t* = −1.133, *p* > 0.05, whereas at higher levels of HPWS (mean + 1 standard deviation), the positive effect of organizational psychological ownership on job crafting was significant, with a simple slope *b* = 0.282, *t* = 3.072, *p* < 0.01. It can be seen that the moderating effect of a high-performance work system between organizational psychological ownership and job crafting is significant, specifically, the positive relationship between organizational psychological ownership and job crafting is significant at higher HPWS. Hypothesis 3c was further validated. The specific cross-level mediating moderating effects are shown in Figure 3.

## 5. Discussion

Procedural justice has a significant impact on employees’ job-crafting behaviors. Procedural justice can be established by ensuring fairness, transparency, and objectivity in decision-making and resources to provide just treatment and opportunities. If an organization performs poorly in terms of procedural justice, it may lead to employee dissatisfaction and mistrust, which in turn affects employees’ job-crafting behaviors. Consistent with most research findings, organizational fairness, especially procedural justice, positively predicts employee job-crafting behavior ([14]). We can explain the relationship between procedural justice and job crafting through the perspectives of social exchange theory and equity theory; when employees feel treated fairly by the organization, employees maintain positive evaluations and trust, embrace, and show obedience to their managers, and increase their sense of dependence on the organization, so that employees feel a sense of being valued and dependent on their work group and are more motivated, willing, and perceive themselves as having a duty to engage in job crafting. Therefore, organizations should focus on and actively maintain procedural justice in order to encourage employees to engage more actively in job-crafting behaviors. Through fair promotion opportunities, compensation systems, training and development opportunities, and transparent decision-making processes, employees can be motivated to contribute to the success of the organization by increasing their satisfaction and loyalty.

Organizational psychological ownership assumes a mediating role between procedural justice and employee job-crafting behavior. This study confirms that procedural justice can influence job-crafting behaviors by increasing employees’ organizational psychological ownership. Integrating job demand-resource theory and social exchange theory helps us to understand the role of organizational psychological ownership in the relationship between procedural justice and job crafting. Consistent with the predictions of previous studies, organizations presenting procedural justice environments can increase employees’ organizational psychological ownership, and the sense of organizational psychological ownership motivates employees to think about tasks, learn relevant task skills, increase employees’ domain expertise, inspire employees to take responsibility for tasks, and assume responsibility for consideration of the organization’s interests and prospects ([16]; [22]), which collectively drive employees to actively optimize their work tasks and approaches in ways that are consistent with their capabilities and contribute to the organization’s development.

High-performance work systems assume a moderating role between organizational psychological ownership and employee job crafting. This study confirms that high-performance work systems can provide employees with work resources when employees perceive that the organization provides them with resources and grants them autonomy, they value the organization more psychological incentives for employees to fully utilize their self-worth and are more likely to engage in job crafting activities ([15]; [2]). This result is consistent with the prediction of self-determination theory, and, therefore, the hypothesis that when high-performance work systems are implemented to a greater extent in firms, it can increase employees’ sense of organizational belonging, which in turn facilitates employees to engage in job crafting is validated. According to the results of previous studies, high-performance work systems, as a management model aimed at improving employee performance and productivity, can enhance employees’ job-crafting behaviors by moderating procedural justice, i.e., employees’ perceptions of the fairness and reasonableness of the organization’s decision-making process ([23]). However, the results of this study indicate that the moderating effect of high-performance work systems on the relationship between procedural justice and employee job crafting is not significant. This is not consistent with previous research and theoretical predictions, which may be due to the fact that high-performance work systems may have multiple moderating effects on the relationship between procedural justice and employee job crafting, and that such moderating effects may vary depending on the type of job and employee characteristics. Overall, however, high-performance work systems have a positive effect on the relationship between organizational psychological ownership and employee job crafting, but their moderating effects may be limited by other factors and interactions. Future research needs to explore the moderating mechanisms of high-performance work systems in greater depth and precisely control for the influencing factors to better understand their effects on the relationship between procedural justice and employee job crafting.

### 5.1. Theoretical Implications

First, while the literature has started to explore the antecedent variables of job crafting, there is still a lack of research on organizational contextual variables ([27]). Our study confirms that organizational procedural justice significantly influences employee job crafting, as supported by social exchange theory and equity theory. Procedurally fair environments within organizations stimulate employee initiative and reinforce job crafting. This finding not only enriches the literature, but also provides a theoretical foundation for organizations to promote job crafting more effectively in practice, which is crucial for both understanding and implementing job crafting.

Second, we reveal how procedural justice promotes employee job crafting, confirming the mediating role of organizational psychological ownership in this relationship. By demonstrating the mediating mechanism of organizational psychological ownership, we contribute to the understanding of how procedural justice influences employees’ proactive behaviors, thus providing a valuable path to uncover the “black box” of this mediating mechanism. Previous studies have suggested that fairness is a key antecedent of organizational psychological ownership, which is closely linked to job crafting ([35]; [17]; [23]). Our findings, based on job demand-resource theory and social exchange theory, offer empirical evidence for these arguments and enhance the comprehension of the mediating mechanisms involved.

Third, this study integrates self-determination theory into the dynamic interactions between employees and organizational environments and explores the role of high-performance work systems in the entire process of employee job crafting. By advancing the study of human resource management practices and introducing HPWS as a moderator, we enrich the research on procedural justice and the boundary conditions of organizational psychological ownership. This study not only extends the application of HPWS in organizational behavior research, but also offers new insights for organizations in designing and implementing high-performance work systems.

### 5.2. Practical Implications

The practical significance of this study can be summarized as follows:Procedural justice can help organizations develop effective monitoring mechanisms to ensure fair decision-making and communication processes. It also empowers employees to voice their opinions in team decision-making, stimulating their work enthusiasm and creativity. This improves employee participation, commitment, and sense of belonging, thereby enhancing job-crafting behavior. Therefore, companies should prioritize the establishment of fair processes.Managers should deeply understand employees’ organizational psychological ownership and analyze its long-term development. Appropriate human resource management measures, such as establishing a fair management system and creating a supportive atmosphere, can increase employees’ psychological ownership, guiding and motivating positive psychological perceptions to transform into desired positive behaviors. This can boost job satisfaction, loyalty, personal growth, and organizational core competitiveness and sustainability.In the context of enhancing organizational psychological ownership, enterprises should recognize the role of high-performance work systems. By implementing effective risk management mechanisms, organizations can identify and assess various risks and take appropriate actions. In an uncertain environment, remaining flexible and adaptable while focusing on employee development and performance management is crucial. These measures can help organizations better leverage their high-performance work systems.

## 6. Limitations and Prospects

First, this study is limited by the research conditions; this study’s use of self-reported data from individuals over a period of time does not allow for valid inferences to be made about the causal relationships between the research variables, nor does it provide strong evidence about causal conclusions. Although this study demonstrates that procedural justice can have an impact on job crafting through organizational psychological ownership, it ignores the time factor in which organizational psychological ownership has an effect, thus taking some time to lead to job crafting. If longitudinal data are used, the causal relationship between procedural justice, organizational psychological ownership, and job crafting can be tracked and explored. Therefore, future research should use tracking data to make causal inferences from existing models. Second, the moderating role of the high-performance work system part of the data was not significant, and we predicted that there might be some errors in which the high-performance work system has multiple moderating roles. In the future, we can try to use a new method to cross-study the HRM system at the organizational level and employee behavior at the individual level to reduce the error of the results. Finally, due to various limitations, this study did not distribute paper questionnaires, and all questionnaires were distributed and collected through the Internet. This may lead to the lack of field research in this study. In future studies, it may be considered to collect data on a larger regional scale and verify the applicability of the findings of this study.

## Figures and Tables

**Figure 1 behavsci-15-00004-f001:**
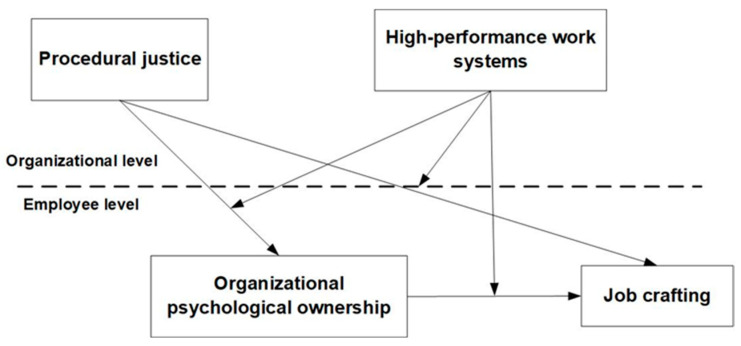
Theoretical model.

**Figure 2 behavsci-15-00004-f002:**
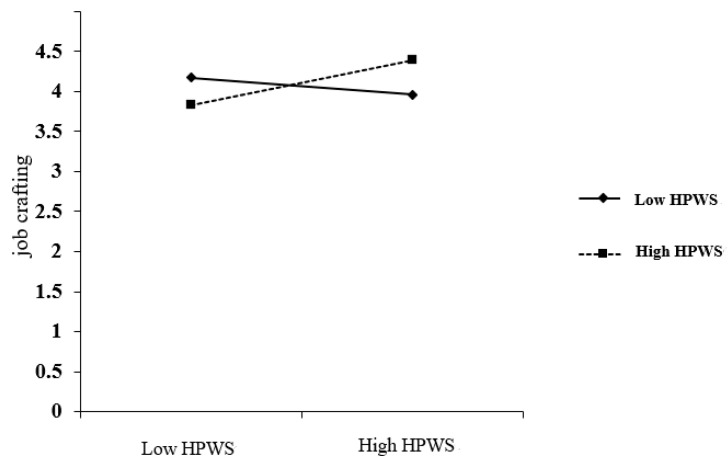
The moderating effect of high-performance work systems (HPWS) on the relationship between procedural justice and organizational psychological ownership.

**Figure 3 behavsci-15-00004-f003:**
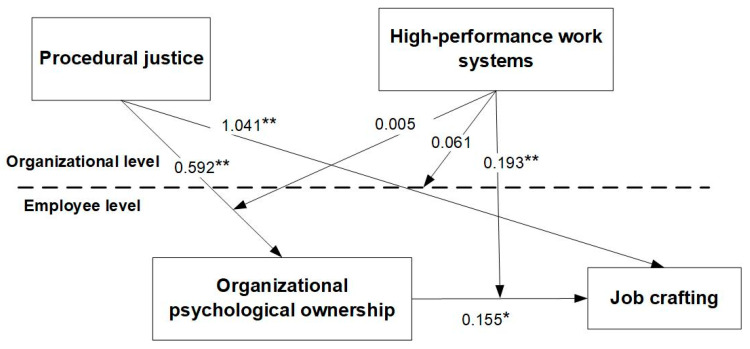
Cross-layer mediated regulation model. Note: * *p* < 0.05; ** *p* < 0.01.

**Table 1 behavsci-15-00004-t001:** Descriptive statistics of basic employee information (N = 1049).

Variant	Form	Number of People	Percentage
Distinguishing between the sexes	Male	478	45.57%
	Female	571	54.43%
(a person’s) Age	25 and under	71	6.77%
	26–30 years	261	24.88%
	31–35 years	404	38.51%
	36–40 years	252	24.02%
	41 and over	61	5.82%
Educational level	Three-year college	197	18.78%
	Undergraduate (adjective)	637	60.72%
	Bachelor’s degree	173	16.49%
	Doctoral	42	4.00%
Length of service	1 year and less	121	11.53%
	2–3 years	376	35.84%
	4–5 years	346	32.98%
	6–10 years	148	14.11%
	11 years and above	58	5.53%
Duties	Technical staff	359	34.22%
	Functional post holders	361	34.41%
	Operational staff	202	19.26%
	Primary manager	70	6.67%
	Middle manager	50	4.77%
	Senior management	7	0.67%

**Table 2 behavsci-15-00004-t002:** Descriptive statistics of basic information of enterprises (N = 76).

Variant	Form	Number of Enterprises	Percentage
Type of business	Private business	61	80.26%
	State-owned/collective enterprises	7	9.21%
	Foreign/Joint venture	6	7.89%
	(sth. or sb) else	2	2.63%
Enterprise size	1–50 persons	8	10.53%
	51–100 persons	21	27.63%
	101–200	27	35.53%
	201–500	16	21.05%
	501 and above	4	5.26%

**Table 3 behavsci-15-00004-t003:** Results of validation factor analysis (N = 403).

Variant	χ^2^/df	RMR	GFI	RMSEA	NNFI	CFI
Procedural justice	1.613	0.013	0.994	0.024	0.996	0.998
Organizational psychological ownership	2.440	0.018	0.993	0.037	0.992	0.996
High-performance work systems	2.356	0.029	0.950	0.078	0.883	0.948
Job crafting	2.741	0.013	0.994	0.041	0.993	0.997

Note: RMR stands for Root Mean Square Residual; GFI stands for Goodness-of-Fit Index; RMSEA stands for Root Mean Square Error of Approximation; NNFI stands for Non-normal of Fit Index; CFI stands for Comparative Fit Index.

**Table 4 behavsci-15-00004-t004:** Descriptive statistics and correlation analysis (N = 403).

Variant	M	SD	1	2	3	4	5	6	7	8	9	10	11	12
1. Distinguishing between the sexes	1.54	0.50												
2. (A person’s) age	2.97	1.00	−0.04	-										
3. Educational level	2.06	0.71	0.06	0.04	-									
4. Length of service	2.66	1.03	−0.02	0.46 **	−0.01	-								
5. Duties	2.15	1.15	−0.04	0.28 **	0.05	0.35 **	-							
6. Type of business	1.41	0.78	0.03	0.14 **	0.03	0.12 **	0.12 **	-						
7. Enterprise size	2.88	1.08	0.00	−0.06 *	−0.03	0.13 **	−0.03	0.07 *	-					
8. Corporate locations	1.44	0.59	−0.05	0.03	0.04	0.02	−0.01	0.02	−0.21 **	-				
9. Procedural justice	3.98	0.80	−0.03	−0.06	0.13 **	0.09 **	−0.15 **	−0.02	0.00	−0.10	**0.899**			
10. Organizational psychology ownership	3.78	0.85	0.01	0.03	0.09 **	0.01	−0.09 **	0.03	−0.01	0.07 *	0.29 **	**0.892**		
11. High-performance work systems	3.96	0.24	0.06	−0.06 *	0.09 **	0.09 **	−0.18 **	−0.09 **	0.17 **	0.04	0.24 **	0.21 **	**0.958**	
12. Job crafting	4.06	0.83	0.00	−0.08 *	−0.17 **	−0.06	−0.13 **	−0.05	−0.02	−0.12 **	0.85 **	0.21 **	0.21 **	**0.875**

Note: Gender: 1 = Male, 2 = Female. Employee’s age: 1 = 25 years old and below, 2 = 26–30 years old, 3 = 31–35 years old, 4 = 36–40 years old, 5 = 41 years old and above. Educational level: 1= College, 2 = Bachelor’s Degree, 3 = Master’s Degree, 4 = Doctoral Degree. Years of working experience: 1 = 1 year and below, 2 = 2–3 years, 3 = 4–5 years, 4 = 6–10 years, 5 = 11 years and above. Position: 1 = Technical staff, 2 = Functional position staff, 3 = Business department staff, 4 = Basic manager, 5 = Middle manager, 6 = Higher manager. Type of enterprise: 1 = Private enterprise, 2 = State-owned/collective enterprise, 3 = Foreign investment/Joint venture, 4 = Other. Enterprise size: 1 = 1–50 people, 2 = 51–100 people, 3 = 101–200 people, 4 = 201–500 people, 5 = 5–11 years and above. Work experience: 1 = 1 and below, 2 = 2–3 years, 3 = 4–5 years, 4 = 6–10 years, 5 = 11 and 201–500 people, 5 = 501 people or more. * *p* < 0.05; ** *p* < 0.01 below. The bolded numbers in the table are the internal consistency reliability coefficients of the corresponding scales.

**Table 5 behavsci-15-00004-t005:** Results of hypothesis testing.

Variant	Model 1 (Direct Effects)		Model 2 (Full Model)			
	Procedural Justice		Organizational Psychology Ownership		Job Crafting	
	Ratio	95% CI	Ratio	95% CI	Ratio	95% CI
**Individual level**						
Distinguishing between the sexes	−0.024	[−0.098, 0.050]	−0.010	[−0.106, 0.086]	−0.015	[−0.087, 0.057]
(A person’s) age	0.013	[−0.026, 0.051]	0.041	[−0.017, 0.098]	0.004	[−0.034, 0.043]
Educational level	−0.014	[−0.089, 0.062]	−0.001	[−0.100, 0.098]	−0.024	[−0.098, 0.050]
Length of service	0.008	[−0.033, 0.050]	0.031	[−0.023, 0.085]	0.002	[−0.038, 0.043]
Organizational psychological ownership					0.155 *	[0.052, 0.257]
**Team Level**						
Procedural justice (aggregation to team level)	1.041 **	[0.976, 1.107]	0.529 **	[0.362, 0.695]	0.873 **	[0.746, 1.000]
High-performance work systems			0.085	[−0.024, 0.194]	0.023	[−0.031, 0.078]
Procedural justice × high-performance work systems	0.061	[0.016, 0.100]	0.005	[−0.167, 0.177]		
Organizational psychology ownership × high-performance work systems					0.193 *	[0.055, 0.332]
**Individual level**						
Distinguishing between the sexes	−0.024	[−0.098, 0.050]	−0.010	[−0.106, 0.086]	−0.015	[−0.087, 0.057]
(A person’s) age	0.013	[−0.026, 0.051]	0.041	[−0.017, 0.098]	0.004	[−0.034, 0.043]
Educational level	−0.014	[−0.089, 0.062]	−0.001	[−0.100, 0.098]	−0.024	[−0.098, 0.050]
Length of service	0.008	[−0.033, 0.050]	0.031	[−0.023, 0.085]	0.002	[−0.038, 0.043]

Note: * *p* < 0.05; ** *p* < 0.01.

## Data Availability

The data presented in this study are available on request from the corresponding authors.

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
