# Peer review of "How Does Procedural Justice Affect Job Crafting? The Role of Organizational Psychological Ownership and High-Performance Work Systems"

_behavsci, 2024, doi:10.3390/bs15010004_

Round 1

Reviewer 1 Report

Comments and Suggestions for Authors

The manuscript is well written. The introduction is good because it describes the context of the topic of procedural justice affect job crafting, identifies a problem, i.e. paucity of research on the antecedent variables of job crafting, particularly the role of situational factors, and then states the purpose of the paper.

The Main Question
The main question addressed by the research is how procedural justice affects job crafting Relevance
I consider the paper relevant to the discourse of job crafting.  The paper addresses a specific gap in the research, which is the paucity of research on the antecedent variables of job crafting, particularly the role of situational factors. Value Addition
The paper builds on the existing research by demonstrating a new pathway for enhancing employee job crafting and by providing practical insights for corporate management. The paper makes a practical suggestion for employers to focus on fostering an environment characterized by procedural justice and promoting organizational psychological ownership. This is good because it brings down the research from being a scholarly endeavor to something practical for employers and human resource practitioners to do.

The methodology used is appropriate and is clearly explained. The results are well presented, and the discussion logically follows the key results and how they compare with previous studies on the subject.

Improvements Methodology:  The methodology is fit for the purpose. The methodology used for data collection is appropriate and is clearly explained. I see no flaws in the methodology. Conclusions: The conclusions flow logically from the findings. The study draws the conclusion that procedural justice significantly and positively impacts employee job crafting, which justifies the place of the study in the body of knowledge concerning job crafting and employee ownership. The conclusion is consistent with the evidence drawn from the findings. Form: Generally, the paper is well written with respect to form and content. However, throughout the paper, spacing between words and parenthesis needs to be checked. Secondly, one of the references needs to be consistent with the rest of the references in the use of capitals and lowercase in titles.   Please see number 17 on the list of references.

There are a couple of minor editorial issues to be addressed. For instance,

Overall, this is a good paper. It addresses a topic that needs more attention because there are not many studies on the subject.

Author Response

For research article

Dear Reviewer,

Thank you very much for your thorough review and the highly positive feedback on our manuscript. We are greatly encouraged by your recognition of our work and are pleased to hear that you found our study valuable and well-executed. We appreciate your commendation on our contributions to the field of job crafting, particularly regarding our emphasis on the organizational contextual perspective and the mediating role of organizational psychological ownership.

We have carefully reviewed and addressed all the minor revisions you suggested.

A point-by-point response to Comments and Suggestions for Authors

Comments 1: The spacing between words and parenthesis needs to be checked.

Response 1:  Thank you for pointing this out. Based on your suggestions, we have carefully examined the entire text and corrected and standardized the formatting and other aspects.

Comments 2: Secondly, one of the references needs to be consistent with the rest of the references in the use of capitals and lowercase in titles.   Please see number 17 on the list of references.

Response 2: Thank you for pointing this out. We have checked all the references and made formatting changes based on your suggestions.

We are committed to refining our manuscript based on your valuable insights. If there are any additional suggestions or areas you feel we could further improve, please do not hesitate to let us know. We are open to all feedback that will enhance the quality of our work.

Once again, thank you for your time and effort in reviewing our manuscript. Your constructive comments have been invaluable in helping us to refine and strengthen our research.

We look forward to your final decision and hope that the revisions we have made meet your expectations.

Best regards,

Reviewer 2 Report

Comments and Suggestions for Authors

I respect the authors for their hard work. However, I have some questions and need further explanations.

1. Introduction

It would be better to describe the differences between this study and previous studies.

2. Theoretical framework

Please add previous research to the theoretical framework

3. Methodology

3.1. Research methodology and sample

How did the researchers select 92 technology-based enterprises and 1,345 employees? What is the period of the survey? Please describe the sampling process in more detail.

4. Discussion

I think the implications are lacking compared to the analysis. In particular, the Practical Implications are lacking, so please supplement them.

* Please present the survey items in the paper. Please present the survey items in 3. Methodology, or 3(4??) Results or Appendix.

Author Response

Dear Reviewer,

Thank you very much for your thorough review and constructive feedback on our manuscript. We greatly appreciate the time and effort you have invested in evaluating our work. Your comments have been very helpful in improving the quality of our research. Below is our detailed response to each of your minor revision suggestions:

Point-by-point response to Comments and Suggestions for Authors

Comments 1: It would be better to describe the differences between this study and previous studies.

Response 1:  Thank you for pointing this out. We added a description of the differences between this study and previous studies to the concluding section of the introduction as suggested. Added content: In summary, this study makes several contributions to job crafting. Firstly, unlike previous studies that focused on outcome variables or individual aspects, we adopt an organizational contextual perspective, emphasizing organizational-level contextual factors. We examine the mediating role of organizational psychological ownership between procedural justice and employees’ job-crafting behaviors using a cross-level model of moderating mediators, and how organizational psychological ownership influences the relationship between job-crafting behaviors under varying levels of high-performance work system implementation.

Comments 2: Please add previous research to the theoretical framework.

Response 2: Thank you for the suggestion that the theoretical framework section, where we are mainly describing the relationships between the variables, may not address prior research on job crafting sufficiently. Therefore, we have added previous research to the section on procedural justice and job crafting as suggested, and the additions are as follows: However, in the past, researchers have mainly focused on the effects of job crafting on the positive psychological cognitive, and affective aspects of individuals (Rudolph et al., 2017), as well as on work attitudes and work outcomes (Kim et al., 2018), for example, employee-authored job crafting behaviors positively affect employees' health (Thun & Bakker, 2018) and job satisfaction (Cheng et al., 2016). Yet less attention has been paid to how organizational contextual factors such as procedural justice, as antecedent variables, affect job crafting.

Comments 3: How did the researchers select 92 technology-based enterprises and 1,345 employees? What is the period of the survey? Please describe the sampling process in more detail.

Response 3: Thank you for pointing this out. We have clarified the methodology section of the study in detail, specifically describing the exact process and timing of sample selection and data collection. This is specified below:

This study employs a questionnaire survey method to examine 92 science and technology-based enterprises in eastern China, encompassing sectors such as new energy, advanced manufacturing, new materials, electronic information, and biotechnology, with involvement from various departments including technology, finance, and HR. We selected science and technology-based enterprises as the research subjects because these enterprises typically exhibit high innovation vitality and technological content, playing a crucial role in dynamic market environments. We screened enterprises that met these criteria by reviewing their registration information, industry reports, and public information. To ensure the sample’s representativeness, we selected enterprises ranging from small to large, with employee counts between 50 and 5,000. We initially screened approximately 200 technology-based enterprises using industry reports and corporate databases. We then contacted the HR departments or management of these enterprises to explain the study’s purpose and methodology and to solicit their willingness to participate. Ultimately, 92 companies agreed to participate in our study. In each participating firm, we selected employees using a random sampling method. At least 10 employees were selected from each company to ensure a diverse and representative sample. In total, we collected survey data from 1,345 employees. The questionnaire survey was conducted from August 1, 2021, to December 31, 2021, spanning three months. Before administering the questionnaire, we obtained permission from the enterprise leaders and the HR department. Employees were fully informed of the survey’s purpose and the confidentiality of the results. Each participant received a small gift to reduce stress after completing the survey. To control for homogeneous variance, this study used a multi-source data collection approach. The questionnaires were coded and numbered for each organization before distribution.HR executives evaluated the high-performance work system and basic firm information, while employees assessed job crafting, organizational psychological ownership, procedural justice, and personal information (e.g., gender, age, education level, tenure, and job title).

Comments 4: I think the implications are lacking compared to the analysis. In particular, the Practical Implications are lacking, so please supplement them.

Response 4: Thank you for your valuable advice. Based on the suggestions, we have added two new sections, theoretical and practical, to the discussion section to fully describe the contributions of this study. The additions are as follows:

4.1 Theoretical Implications

First, while the literature has started to explore the antecedent variables of job crafting, there is still a lack of research on organizational contextual variables (Park & Park, 2021). Our study confirms that organizational procedural justice significantly influences employee job crafting, as supported by social exchange theory and equity theory. Procedurally fair environments within organizations stimulate employee initiative and reinforce job crafting. This finding not only enriches the literature but also provides a theoretical foundation for organizations to promote job crafting more effectively in practice, which is crucial for both understanding and implementing job crafting.

Second, we reveal how procedural justice promotes employee job crafting, confirming the mediating role of organizational psychological ownership in this relationship. By demonstrating the mediating mechanism of organizational psychological ownership, we contribute to the understanding of how procedural justice influences employees proactive behaviors, thus providing a valuable path to uncover the black box of this mediating mechanism. Previous studies have suggested that fairness is a key antecedent of organizational psychological ownership, which is closely linked to job crafting (Wang et al., 2018; K & Ranjit, 2021; Mehmood et al., 2021). Our findings, based on job demand-resource theory and social exchange theory, offer empirical evidence for these arguments and enhance the comprehension of the mediating mechanisms involved.

Third, this study integrates self-determination theory into the dynamic interactions between employees and organizational environments and explores the role of high-performance work systems in the entire process of employee job crafting. By advancing the study of human resource management practices and introducing HPWS as a moderator, we enrich the research on procedural justice and the boundary conditions of organizational psychological ownership. This study not only extends the application of HPWS in organizational behavior research but also offers new insights for organizations in designing and implementing high-performance work systems.

4.2 Practical Implications

The practical significance of this study can be summarized as follows:

1. Procedural justice can help organizations develop effective monitoring mechanisms to ensure fair decision-making and communication processes. It also empowers employees to voice their opinions in team decision-making, stimulating their work enthusiasm and creativity. This improves employee participation, commitment, and sense of belonging, thereby enhancing job-crafting behavior. Therefore, companies should prioritize the establishment of fair processes.

2. Managers should deeply understand employees’ organizational psychological ownership and analyze its long-term development. Appropriate human resource management measures, such as establishing a fair management system and creating a supportive atmosphere, can increase employees’ psychological ownership, guiding and motivating positive psychological perceptions to transform into desired positive behaviors. This can boost job satisfaction, loyalty, personal growth, and organizational core competitiveness and sustainability.

3. In the context of enhancing organizational psychological ownership, enterprises should recognize the role of high-performance work systems. By implementing effective risk management mechanisms, organizations can identify and assess various risks, and take appropriate actions. In an uncertain environment, remaining flexible and adaptable while focusing on employee development and performance management is crucial. These measures can help organizations better leverage their high-performance work systems.

Comments 5: Please present the survey items in the paper. Please present the survey items in 3. Methodology, or 3(4??) Results or Appendix.

Response 5: Thank you for pointing this out. We have submitted the relevant questionnaires as Appendix in both original and translated versions.

Thank you again for your valuable comments and suggestions. We are confident that these changes will make our article better and more rigorous. Please feel free to let us know if you have any further suggestions or need us to make more adjustments.

Looking forward to your final decision.

Best regards,
